# Effect of nitrogen limitation and soil biophysics on Holocene greening of the Sahara

Jooyeop Lee[1], Martin Claussen[2,3], Jeongwon Kim[1], Je-Woo Hong[4], In-Sun Song[5], and Jinkyu Hong[1]

[1]Ecosystem-Atmosphere Process Laboratory, Department of Atmospheric Sciences, Yonsei University, Seoul, Korea (Republic of)
[2]Max-Planck-Institut für Meteorologie, Hamburg, Germany
[3]Center for Earth System Research and Sustainability (CEN), Universität Hamburg, Germany
[4]Korea Adaptation Center for Climate Change, Korea Environment Institute, Sejong, Korea (Republic of)
[5]Mathematical Atmospheric Physics Lab, Department of Atmospheric Sciences, Yonsei University, Seoul, Republic of)

*Correspondence to*: Jinkyu Hong (jhong@yonsei.ac.kr)

**Abstract.** The so–called Green Sahara (GS), wet and vegetative Sahara region in the early to mid–Holocene, provides useful information on our climate simulation because it is a consequence of complex interaction between biophysical and climatic processes. It is still a challenge to simulate the GS in terms of vegetative extent and precipitation using the current climate models. This study attempts to simulate the 8,000 year ago Green Sahara by using the state–of–the–art Earth system model CESM that incorporates the nitrogen cycle and the soil–precipitation feedbacks. Our study puts more emphasis on the impact of soil biophysical properties (e.g., bare-soil albedo, porosity, heat capacity, and hydraulic conductivity) and soil nitrogen influenced by soil organic matter on the simulation of the GS. In this coupled simulation, vegetation interacts with changes in soil properties and soil organic matter by phenology, decomposition and allocation of carbon and nitrogen. With changes in the Earth's orbit and dust in the early to mid–Holocene, the model simulates increased precipitation in North Africa, but does not capture the extent of the GS. Our analysis shows that the Holocene greening is simulated better if the amount of soil nitrogen and soil texture are properly modified for the humid and vegetative GS period. Soil biochemical and physical properties increase precipitation and vegetation cover in North Africa through their influence on photosynthesis and surface albedo and their consequent enhanced albedo– and evapotranspiration–precipitation feedbacks. Our findings suggest that future climate simulation needs to consider consequent changes in soil nitrogen and texture with changes in vegetation cover and density for proper climate simulations.

## 1. Introduction

Sahara is the largest subtropical desert on Earth, but it was wetter and had more vegetation than today during the early to mid–Holocene (EMH hereafter) (Holmes, 2008). In this Green Sahara (GS hereafter) period, the region was covered with various vegetation, inland rivers and mega–lakes (Pachur and Kröpelin, 1987; Jolly et al., 1998; Schuster et al., 2005; Quade et al.,

2018) up to 23° N of North Africa, unlike the current vegetation which only exists in regions below 15° N. The GS was sensitive to climate change with locally rapid transitions from a humid to a more arid state some 8,000 to 5,000 years ago (Shanahan et al., 2015). The GS and its transition to an arid state are the results of complex interactions of the orbital forcing changes with the land–atmosphere interaction, and sea surface temperature (SST) variability (Claussen et al. 2017; Braconnot et al., 2019). Future climate predictions are made using Earth system models (ESMs), and recent progress in the development of ESMs made it possible to simulate and to assess the carbon–climate feedbacks. GS is a good example to evaluate various feedback processes in the state-of-the-art ESMs and to evaluate the model performance accordingly (Harrison et al., 2015; Pausata et al., 2020). In this context, the GS period provides unique and useful insights for our climate prediction in a changing climate.

Many modeling studies have tried to simulate the GS and to understand the underlying mechanisms of the GS correctly. Early studies focused on the impacts of change in the Earth's orbit leading to higher summer solar insolation compared to the present (Kutzbach et al., 1981; Kutzbach and Street-Perrott, 1985). Further studies revealed that the following factors contributed to the amplification of the GS; the ocean–climate feedback (Kutzbach and Liu, 1997; Braconnot et al., 1999), vegetation–climate feedback (Claussen, 1998; Claussen et al., 1999; Claussen et al., 2013; Rachmayani et al., 2015; Groner et al., 2018), soil–climate feedback (Kutzbach et al., 1996; Knorr et al., 2001; Levis et al., 2004; Knorr et al., 2006; Vamborg et al., 2011; Lu et al., 2018), inland water (Coe and Bonan, 1997; Krinner et al., 2012), and dust–climate feedback (Pausata et al., 2016; Gaetani et al., 2017). However, current state–of–the–art climate models still yield diverging results regarding the extent of the GS and the understanding of feedback processes (Claussen et al., 2017).

Recently, ESMs began to couple nitrogen (N hereafter) processes to carbon cycle. The global carbon budget has been simulated more realistically with the recent inclusion of the nitrogen cycle in the ESM because of the N–limitation effect of terrestrial gross primary production (GPP) (Thornton et al., 2007; Castillo et al., 2012; Arora et al., 2020). However, impact of nitrogen limitation on the GS simulation has not been extensively investigated as far as we know. It is also notable that soil in the GS includes more organic matter and humus compared to the current condition because vegetation leads to more organic matter in soils, thus affecting hydraulic, thermal, and radiative properties of soil (e.g., Kutzbach et al., 1996; Levis et al., 2004; Lu et al., 2018). Lu et al. (2018) concluded that organic matter in soil texture likely played an important role in the GS dynamics, but their study was based on the offline land surface model only with changes in soil hydraulic and thermal properties and more studies need to be done to investigate impacts of the biophysical and biogeochemical changes in soil on the GS greening in the framework of the fully coupled ESMs.

The present study focuses on better understanding of the ESM's simulation ability to simulate vegetation in the GS and its uncertainties by the representation of physical processes in the land surface model. We investigate the impacts of nitrogen limitation and soil biophysical processes on the GS simulations using the Community Earth System Model (CESM, version 1.2) (Hurrell et al., 2013). Based on a series of the model simulations, we examine changes in the GS extension and its associated uncertainties by related factors of orbital forcing, nitrogen and carbon in soil, and soil texture.

## 2. Methods

### 2.1 Model description

The Community Earth System Model (CESM, version 1.2) with the Community Atmosphere Model version 4 (CAM4; Neale et al., 2013) and Community Land Model version 4 (CLM4; Oleson et al., 2010; Lawrence et al., 2011) is employed in this study. The CESM incorporates prognostic carbon–nitrogen (CN) equations and the dynamic vegetation model into the land surface model. The model simulates carbon dynamics such as productivity, phenology, decomposition and allocation with carbon and nitrogen fluxes at the ecosystem-atmosphere interface by the CN prognostic equations. This new CN parameterization makes a substantial difference in vegetation simulation by limiting gross primary production ($GPP$) with soil mineral nitrogen in the climate simulations particularly (Thornton et al., 2007; Castillo et al., 2012; Arora et al., 2020):

$$GPP = GPP_p \cdot \left(1 - \frac{CF_{excess}}{GPP_p}\right) \tag{1}$$

$GPP$ becomes the potential $GPP$ ($GPP_p$) if there is no nitrogen limitation and nitrogen limitation is linked to soil organic matter in the model. Nitrogen limitation on $GPP$ is expressed by $CF_{excess}$ which is decided by soil mineral nitrogen ($N_{sminn}$). $N_{sminn}$ is a function of the conversion rate of soil organic nitrogen to mineral nitrogen ($NF_{soil4n \to sminn}$) and is affected by the amount of soil carbon ($soil4c$) eventually (Eq. (2) – (9)). It was reported that $GPP$ in CESM version 1 was smaller than other CMIP6 models because of the larger nitrogen limitation effect. More details on the CN parameterization in the CLM can be found in Thornton et al. (2002), Thornton et al. (2009), and Kluzek (2012). The description of the variables used in this study including those in Equation 1 are listed in Appendix A.

$$CF_{excess} = CF_{avail\_alloc} - CF_{alloc} \tag{2}$$

$$CF_{avail\_alloc} = GPP_p - MR \tag{3}$$

$$CF_{alloc} = \left(N_{plant\_demand} \cdot f_{pg} + N_{retrans}\right) \cdot \frac{k_{c\_alloc}}{k_{n\_alloc}} \tag{4}$$

$$f_{pg} = \frac{N_{uptake}}{N_{plant\_demand}} \tag{5}$$

$$N_{plant\_demand} = CF_{avail\_alloc} \cdot \frac{k_{n\_alloc}}{k_{c\_alloc}} - N_{retrans} \tag{6}$$

$$N_{uptake} = \frac{N_{sminn}}{\Delta t} - NF_{immob} \tag{7}$$

$$N_{sminn} = N_{sminn} + NF_{soil4n \to sminn} * dt \tag{8}$$

$$NF_{soil4n \to sminn} = \frac{soil4c * k_{decomp}}{dt * 10} \tag{9}$$

The albedo–precipitation feedback has been also considered to be one of the important aspects in the land–atmosphere interaction and should be clearly examined in the GS simulation (e.g., Charney, 1975; Cess, 1978; Bonfils et al., 2001; Houldcroft et al., 2009; Levine and Boos, 2017). Precipitation increases if surface albedo decreases because of more vegetation through the stratification of the atmosphere particularly in drylands. However, in the CMIP5 (Coupled Model Intercomparison Project 5) EMH simulations, soil moisture is irrelevant to surface albedo in most land surface models (LSMs) (e.g., Takata et al., 2003; Zeng, 2005; Houldcroft et al., 2009; Oleson et al, 2010; Watanabe et al., 2010; Vamborg et al., 2011). This study uses CLM4 for simulation of land surface processes. In the CLM4, surface albedo is parameterized as a function of soil moisture in the calculation of soil background albedo. In this parameterization, soil background albedo decreases with increases in soil moisture of the first soil layer as follows:

$$\alpha_{soil} = \alpha_{sat} + \Delta \tag{8}$$

$$\Delta = \begin{cases} 0.11 - 0.40\theta_1 \ (\theta_1 \leq 0.275) \\ \qquad 0 \ (\theta_1 > 0.275) \end{cases} \tag{9}$$

In addition, the soil texture distribution is classified by sand and clay percentage and soil has 20 color classes from the low to the high albedo. In the lowest albedo (darkest soil), dry (saturated) albedo is 0.08 (0.04) and 0.16 (0.08) in visible and NIR range, respectively. In the highest albedo (lightest soil), the dry (saturated) albedo is 0.36 (0.25) and 0.61 (0.50) in visible and NIR range, respectively. $\alpha_{sat}$ is equal to the albedo of the saturated soil.

## 2.2 Experimental design

For our CESM simulations of the GS, the horizontal resolution is 4°×5° in global scale with 26 vertical levels. The initial land surface data are made using CLM4 offline simulations for 600 years in the CN accelerated decomposition mode (Thornton and Rosenbloom, 2005), and then using 1200–year CLM4 simulations with the CNDV (Carbon–Nitrogen–Dynamic–Vegetation) with the atmospheric forcing data of 1948–1972 years based on Qian et al. (2006). The present (i.e., 0K) and EMH (i.e., 8K) simulations are 500–year climate simulations using the pre–industrial and EMH conditions, respectively (Table 1). The EMH period is set to 8000 years ago (8K) in this study because 8K period was relatively greener than other periods (Jolly et al., 1998; Groner et al., 2018).

Additional sensitivity experiments using the EMH conditions are conducted with different land states to examine the impacts of soil nitrogen, soil texture, and the mega lake on the GS simulation (Table 1). Soil carbon and nitrogen are negligible in the current Sahara. To the best of our knowledge, there is no information on soil organic carbon and nitrogen in North Africa during the EMH. It is reasonable to consider more soil carbon and nitrogen in the GS than in the present–day Sahara, because of their close coupling with biomass (Brooks, 2003; Wang et al., 2010; Yang et al., 2014). To consider the possibility of relatively larger soil carbon and nitrogen in the GS, more soil carbon and nitrogen are added with less sandy soil (i.e., loamier soil with more soil organic matter) in the 8KCN and 8KCNS experiments based on Jolly et al. (1998) to prescribe soil organic

matter in the GS (i.e., 10–22° N, 345° W–40° E) using nearby grassland grid values (Fig. 1). In these sensitivity experiments, the feedbacks between vegetation and increased soil carbon and nitrogen evolve dynamically in the model through litter–fall and decomposition processes.

In all EMH simulations, the Earth's orbital parameters and atmospheric $CO_2$ concentration are prescribed based on Berger (1978) with pre-industrial SST (Table 2), and atmospheric dust is reduced following Pausata et al. (2016) (Fig. 2). In addition, the Lake Mega–Chad is considered in our EMH simulations by assigning 73% of the two grids near the Lake Chad region (14–18° N, 15° E) as lake area to consider the extensive Lake Chad in the EMH (Leblanc et al., 2006; Quade et al., 2018). Our analysis focuses on three critical regions in North Africa around the border of the vegetation in our simulation: the North Africa of 20° N (solid boxed area in Fig. 3, NA hereafter) and the Sahara-Sahel (SS hereafter) region. These regions are selected as the analysis region because the NA region is closely related to the onset and progress of the African Monsoon (Levis et al., 2004) and the SS region was sensitive to precipitation–land interaction (Kutzbach et al., 1996). The Sahara-Sahel region is divided into the Northern Sahara–Sahel region (dotted box area in Fig. 3, NSS hereafter) and Southern Sahara–Sahel region (dashed box area in Fig. 3, SSS hereafter) with a common latitudinal band around 18° N region in this study. The NSS is the boundary region of vegetation in the EMH and the SSS shows substantial increases in precipitation to changes in the orbital forcings (more information in the next section). These two regions share 18° N latitude band for considering the common properties between two regions and the model resolution.

## 3. Results and Discussion

### 3.1 Climate and vegetation simulation in the Green Sahara

The 8K simulation shows typical climate patterns in the Sahara–Sahel region during the EMH and there are substantial differences of surface climates between the 0K and 8K experiments in North Africa (Fig. 3). The summer downward radiation at the top of the atmosphere increases by 6% in the SS region compared to the present because of the change in the orbital forcing and less amount of dust in the EMH. This increase of solar radiation leads to an increase in surface shortwave radiation and net radiation by 30 and 21 W m$^{-2}$, respectively and subsequent temperature warming by 1.8° C in the NA region in the 8K experiment (Fig. 3). The intensified land–sea thermal contrast during the EMH makes spatial changes in precipitation and wind pattern over Holocene North Africa compared to the present climate. Notably, the increases in air temperature and shortwave radiation are not substantial in the SS region (less than 10% compared to the NA region), because of the increase in precipitation and clouds with the northward migration of the monsoon by the intensified land-sea thermal contrast (Fig. 3d). Indeed, the meridional wind in the 18° N region changed from northerly (-0.7 ms$^{-1}$) to southerly (0.3 ms$^{-1}$) in the 8K experiment (Fig. 3g, 3h), indicating a northward shift of the Intertropical Convergence Zone (ITCZ) (i.e., the convergence zone of the trade winds) to above 18° N. This ITCZ shift caused a favorable condition for a moister Sahara by transporting more moisture

and subsequent more precipitation to the SS region (Chikira et al., 2006; Larrasoaña et al., 2013). Compared to the pre–industrial simulation (i.e., 0K), summer precipitation in the EMH simulation (i.e., 8K) increases by 18% (from 4.0 to 4.7 mm day$^{-1}$) and 15% (from 2.5 to 2.9 mm day$^{-1}$) in the SSS and NSS region, respectively (Fig. 3e, 3f).

The pre–industrial simulation captures the observed spatial extent of the current Sahara based on MODIS (Moderate Resolution Imaging Spectroradiometer) land surface data (Fig. 4a) (Broxton et al., 2014). Importantly, the model does not simulate increased vegetation fraction, despite the increased precipitation over the SS region during the EMH (Fig. 4b). Reconstruction of vegetation based on the pollen records reveals that grass and trees expanded up to 26° N and 20° N over North Africa during the GS period, respectively (Hely et al., 2014; Hopcroft et al., 2017). Also, leaf wax data indicates that the GS extended more as far north as 31° N (Tierney et al., 2017). In contrast to the proxy data, vegetation fraction and GPP shows negligible changes in North Africa and even a decrease in western Africa and the southern border of the current Sahara in the 8K simulation (Fig. 4b, 4e). It is notable that Lake Mega–Chad did not make substantial changes in simulated summer precipitation northward movement of the 200 and 412 mm year$^{-1}$ precipitation isohyet for 20 % grassland cover, which is similar to Contoux et al. (2013) (Fig. S1, S2).

Hopcroft et al. (2017) attributed such underestimation of the GS to excessive precipitation requirement for vegetation growth in the model (Table 3). Our investigation also shows that the minimum amount of precipitation for vegetation growth is 577 mm year$^{-1}$ in the 8K experiment and this precipitation threshold is larger than the observation value of 281 mm year$^{-1}$ (Hopcroft et al., 2017). This larger threshold indicates that our model has relatively little vegetation cover with respect to the same amount of precipitation and is not fully understood yet in the ESM (Hopcroft et al., 2017). Our investigation reveals that this excess water requirement by vegetation growth becomes smaller with improved water use efficiency (WUE) with proper assignment of soil organic matter and soil texture, which is discussed in the next section.

**3.2 Impact of soil nitrogen on the Green Sahara simulation**

The 8KCN experiment is designed to study the effect of an increase in soil nitrogen by vegetation on the GS and simulates typical surface climate patterns during the EMH (not shown here). Importantly, the 8KCN simulation reproduces larger precipitation and vegetative areas with an increase in GPP in the SS region, compared to the 8K experiment (Fig. 4). Vegetation fraction increases about twice in the SS region (21 % and 8 % in the SSS and NSS regions, respectively) in the 8KCN experiment (Table 4). The 8KCN experiment also shows 2 times larger vegetation fraction than the 8K experiment around the boundary of precipitation changes between the present and EMH (about 18° N). Particularly, the 8KCN experiment shows 3.3 and 5.3 times larger GPP in the NSS and SSS regions, respectively, compared to the 8K experiment (Table 5). Consequently, the precipitation threshold for vegetation decreases to 412 mm year$^{-1}$ in the 8KCN from 577 mm year$^{-1}$ in the 8K experiment together with increased WUE values (Fig. 5 and 6). With this increase in vegetation fraction, the desert–grassland boundary moves one grid box northward in the 8KCN simulation compared to the 8K experiment (Fig. 4i).

Our analysis shows that this increased vegetation in the 8KCN results from moderated nitrogen–limitation on photosynthetic carbon uptake (Eq. (1) and Fig. 7). The nitrogen downregulation fraction defined by $GPP/GPP_p$ is about 0.5 and 0.3 in the NSS and SSS regions, respectively in the 8K experiment but decreases by about 0.1 in the 8KCN experiment. Our further simulation indicates that vegetation fraction and vegetation types depend on the amount of soil nitrogen (Fig. S3). The increase in vegetation fraction and tree cover fraction are approximately proportional to the amount of soil nitrogen. Vegetation fraction increase is substantially smaller in the 8KCNh (i.e., half nitrogen simulation of the 8KCN) compared to the 8KCN (Fig. S3). Trees in the 8KCN is also substantially replaced by grass and shrub in the simulation with less soil nitrogen (8KCNh).

We speculate that albedo–precipitation feedback is reinforced with more vegetation in the 8KCN experiment. In the 8K experiment, the surface albedo decreases compared to the 0K experiment, mainly because of the increase in soil moisture by the enhanced precipitation. In the 8KCN simulation, the surface albedo diminishes more, because the vegetation increases more, and the surface albedo decreases with increasing vegetation cover. Our further analysis suggests that such feedback reinforcement makes an additional 0.1 mm day$^{-1}$ precipitation and expansion of vegetation, especially in the SSS region (Fig. 8a and 8b).

### 3.3 Impact of soil texture on the Green Sahara simulation

Soil is classified as sandy soil for the present–day Sahara in the 8K and 8KCN simulations like previous studies on the GS (e.g., Hopcroft et al., 2017; Groner et al., 2018). It is, however, plausible that soil in the GS includes more nutrients and humus than sandy soil (i.e., loamier) with the extensive vegetations in this humid region because soil organic matter is formed by decomposition of litter and woody debris (Levis et al., 2004; Koven et al., 2013). Such soil type changes make impacts on soil color as well as soil hydraulic and thermal properties. Compared to sandy soil, loamy soil has larger soil porosity and heat capacity but smaller saturated hydraulic conductivity because they contain more slit and clay than sandy soils. Consequently, loamy soils have larger water-holding (i.e., soil porosity) and heat capacities (Kutzbach et al., 1996; Levis et al., 2004). Previous study based on the offline dynamic vegetation simulation of the GS shows that the simulated greening of the Sahara is sensitive to changes in soil thermal and hydraulic properties without bare-soil albedo changes (Lu et al., 2018).

Our coupled ESM simulation of 8KCNS also shows that soil texture leads to significant changes in vegetation and climate in North Africa. Vegetation cover fraction shows 78% increase over the SS region when the soil nitrogen and soil texture are modified all together (i.e., 8KCNS experiment), compared to the 8KCN experiment. (Fig. 9e; Table 4). Notably, those feedback processes are critical in the western SSS which is coincident with the region is classified as sandy soil in the ESMs (Fig. 9f–9l). Vegetation fraction increase is clearly observed especially in the western SSS (< 10° E) in the 8KCNS experiment, with a 2° N northward movement (15° N to 17° N) of the 20% vegetation coverage border and its related GPP increases. This result suggests that the western SSS is a hot spot in perspective of soil texture in the EMH simulations.

Our analysis indicate that such substantial sensitivity is mainly related to changes in soil hydraulic and thermal properties but the darker soil color of loamy soils also regulates climate and vegetation the study region. Relatively darker soil color of loamy soils in the 8KCNS experiment leads to smaller surface albedo and its increase in reinforced by increases in summertime

precipitation (see Eq. (9); Fig. 9a- 9b). That is, the darker soils enhance absorption of solar radiation (i.e., increase in surface net radiation), and more available energy is partitioned into latent heat fluxes (i.e., smaller Bowen ratio), leading to more precipitation in North Africa (Fig. 9c–9e) possibly through the albedo–precipitation feedback and its related moisture recycling (Hopcroft et al., 2017; Levine and Boos, 2017; Groner et al., 2018). Also, previous studies indicate that enhanced evapotranspiration results in more rainfall by weakened African Easterly Jet (AEJ) (Cook, 1999; Wu et al., 2009; Rachmayani et al., 2015). Our simulation also reveals that AEJ is weakened with more evaporation and summertime rainfall, which is not found in the 8KCN experiment.

## 4. Summary and conclusion

The Green Sahara indicates extensive vegetative area and humid climate in the Sahara-Sahel region in the early to mid-Holocene. Previous studies report that the GS is a consequence of complex interactions between various climatic and biophysical factors. It is also known that land-atmosphere interaction enhances vegetation in the GS period with impacts of SST and orbital forcing change (e.g., Claussen et al., 2013; Groner et al., 2018). This study aims at examining the effect of soil nitrogen and soil physical properties on simulations of the Green Sahara 8,000 years ago using the state-of-the-art ESM that incorporates soil nitrogen and albedo-soil moisture feedback processes. As far as we know, this is the first study to investigate impacts of soil nitrogen and soil texture on the GS simulation using the coupled ESM.

Our ESM simulation results clearly show that vegetation in the GS is dramatically sensitive to soil organic carbon and soil texture in North Africa (Table 5). With the changes in the Earth's orbit and dust in the EMH, the model simulates the northward movement of the tropical rainfall belt and its related precipitation increases in North Africa with the EMH conditions and present–day soil nitrogen and soil carbon. However, despite this more precipitation, the meridional extent of vegetation in the Sahara–Sahel transition zone only marginally differs between the early to mid–Holocene and present–day simulation, which did not simulate greening in the early to mid–Holocene properly. This inconsistent modeling behavior between precipitation and vegetation cover can be attributed to the unrealistically high threshold of precipitation required for desert–grassland transition and low WUE in the terrestrial ecosystem model. Our analysis shows that this excessive rainfall need for vegetation is lowered if enhanced soil nitrogen and loamier soil are considered in the model based on the extensive vegetation during the GS. This change in the vegetation threshold leads to increases in GPP and vegetation cover eventually. Particularly, the SS region is sensitive to soil nitrogen and clay and silt composition in soil (i.e., loamier) in our simulations. Our results suggest that soil property is as important as nitrogen fertilization in simulating precipitation, vegetation dynamics, and atmospheric circulation in North Africa during the EMH.

It is also found that the increase in vegetation cover is reinforced by more precipitation in this soil–nitrogen–enhanced simulation through the feedback in soil albedo, water holding capacity, soil color, and precipitation processes. Our sensitivity simulations show that loamy soil having more soil organic matter also leads to more precipitation and vegetation in the

Holocene North Africa through albedo–precipitation and evapotranspiration–precipitation feedbacks with changes in soil color, soil nitrogen, and soil physical properties. Our results indicate that soil nitrogen through downregulation of GPP and process–based dynamics of soil properties with vegetation are critical for the Green Sahara simulations in the ESMs. In particular, GPP and vegetation cover shows dramatic increases of 400% and 150 % in the SS region respectively with changes in soil organic matter and soil texture. Our findings and their implications can be extended to the future climate and dynamic vegetation simulations. More future study should be paid to the role of soil biogeochemical processes in the vegetation dynamics in climate simulations particularly if we use the ESM to incorporate carbon and nitrogen cycles into the model. Despite our important findings on impacts of soil properties on the GS simulation, further study on the vegetation-soil interactions and its related unparameterized processes such as soil migration is necessary because our modeling studies still have unsatisfactory representation of vegetation cover and WUE notwithstanding ample precipitation in the North Africa.

**Data availability**

All data are available (https://doi.org/10.5281/zenodo5788013) and upon request to the corresponding author (jhong@yonsei.ac.kr / https://eapl.yonsei.ac.kr, last access: 20 February 2021).

**Acknowledgements**

This research has been supported by the National Research Foundation of Korea grant funded by the South Korean government (MSIT) (grant no. NRF-2018R1A5A1024958) and the Korea Meteorological Administration Research and Development Program under Grant KMI2021-01610.

# Appendices

## A. List of symbols and definitions.

| Symbols | Definitions |
|---|---|
| $CF_{alloc}$ | amount of carbon allocated to new growth (gC m$^{-2}$ s$^{-1}$) |
| $CF_{avail\_allloc}$ | amount of total carbon available for new growth allocation (gC m$^{-2}$ s$^{-1}$) |
| $CF_{excess}$ | amount of carbon that plants could additionally allocate if nitrogen were not limiting (gC m$^{-2}$ s$^{-1}$) |
| dt | decomposition timestep (seconds) |
| $f_{pg}$ | fraction of potential growth that can be achieved with the nitrogen available to plants (no units) |
| $GPP_p$ | gross primary production when there is no nitrogen limitation (gC m$^{-2}$ s$^{-1}$) |
| $k_{c\_alloc}/k_{n\_alloc}$ | Carbon:Nitrogen stoichiometry constant for new growth allocation (gC (gN)$^{-1}$) |
| MR | maintenance respiration (gC m$^{-2}$ s$^{-1}$) |
| $N_{plant\_ndemand}$ | nitrogen demand by plant for photosynthesis (gN m$^{-2}$ s$^{-1}$) |
| $N_{retrans}$ | retranslocated nitrogen pool (gN m$^{-2}$ s$^{-1}$). Nitrogen retranslocation is the amount of nitrogen that is removed from the tissues and is stored in the other part of the plant to reuse it in the next growth. |
| $N_{sminn}$ | the amount of mineral nitrogen in the total soil column (gN m$^{-2}$) |
| $N_{uptake}$ | amount of plant uptake from soil mineral nitrogen pool (gN m$^{-2}$ s$^{-1}$) |
| $NF_{immob}$ | amount of immobilized nitrogen flux (gN m$^{-2}$ s$^{-1}$) |
| $NF_{soil4n \rightarrow sminn}$ | nitrogen fluxes leaving soil nitrogen pool with largest turnover time to soil mineral nitrogen (gN m$^{-2}$ s$^{-1}$) |
| soil4c | soil carbon pool with largest turnover time (gC m$^{-2}$) |
| $k_{decomp}$ | decomposition rate constant of the soil organic matter (no unit) |

| | |
|---|---|
| $\alpha_{\text{sat}}$ | soil background albedo for saturated soil (fraction) |
| $\alpha_{\text{soil}}$ | soil background albedo (fraction) |
| $\Delta$ | variable that depends on the volumetric water content of the first soil layer |
| $\theta_1$ | the volumetric soil water content of the first soil layer ($m^3\ m^{-3}$) |

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

Table 1: The abbreviations of the numerical experiments conducted in this study.

| Abbreviation | Description |
|---|---|
| 0K | 500 years simulation using the pre−industrial conditions for orbital parameters and $CO_2$ concentration (284.725 ppm). |
| 8K | 500 years simulation using the orbital parameters and $CO_2$ concentration at 8,000 years ago (Table 2). |
| 8KCN | Same as 8K except that soil carbon and nitrogen are prescribed as the values in the current Sahel region. |
| 8KCN0L | Same as 8KCN except that the Lake Mega–Chad is not considered in the simulation. |
| 8KCNh | Same as 8KCN except that soil carbon and nitrogen are prescribed as half of the 8KCN simulation. |
| 8KCNS | Same as 8KCN except that the North African soil was prescribed as loam. |

Table 2: Orbital parameters prescribed for the simulations.

| Simulation | $CO_2$ concentration | Eccentricity | Obliquity | Longitude of Perihelion |
|---|---|---|---|---|
| 0K | 284.725 ppm | 0.016704 | 23.44 | 283.01° |
| 8K, 8KCN, 8KCN0L,8KCNh, 8KCNS | 259.9 ppm | 0.019101 | 24.209 | 148.58° |

Table 3: 20% vegetation coverage transition rainfall for OBS (Observation data) and 4 model simulations (the Joint UK Land Environment Simulator version 4.1, Lund-Potsdam-Jena dynamic vegetation model version 2.1, Sheffield Dynamic Global Vegetation Model, and CLM4 for this study). OBS and JULES, LPJ, SDGVM indicates the value from the observation and model simulation in the pre–industrial conditions which is from Hopcroft et al., 2017.

| Model name | 20% vegetation coverage transition rainfall (mm yr$^{-1}$) |
|---|---|
| SDGVM | 182 |
| OBS | 281 |
| JULES | 378 |
| LPJ | 515 |
| This study | 570 (0K)<br>577 (8K)<br>412 (8KCN) |

Table 4: Vegetation fraction (%) and GPP (gC m$^{-2}$ month$^{-1}$) in the SS region. Values in parenthesis are percentage changes in vegetation fraction and GPP to the 0K simulation.

| | 0K | 8K | 8KCN | 8KCNS |
|---|---|---|---|---|
| Vegetation fraction (percentage change in vegetation fraction to the 0K simulation) | 8.3 (0%) | 8.3 | 15.0 (81%) | 20.9 (152%) |
| GPP (percentage change in GPP to the 0K simulation) | 3.0 (20%) | 2.5 | 10.3 (312%) | 12.6 (404%) |

Table 5: GPP (gC m$^{-2}$ month$^{-1}$) from 5 CMIP5 models (IPSL-CM5A-LR, MIROC-ESM, bcc-csm1-1, MPI-ESM-P, CCSM4) for 6K simulations and this study in NSS region (18–22 °N) during the pre–industrial period (PI) and early to mid–Holocene (EMH).

| Model name | PI GPP | EMH GPP |
|---|---|---|
| IPSL-CM5A-LR | 0.00 | 0.01 |
| MIROC-ESM | 0.06 | 2.69 |
| CCSM4 | 0.13 | 0.50 |
| bcc-csm1-1 | 1.47 | 1.65 |
| MPI-ESM-P | 1.62 | 5.21 |
| This study | 2.12 (0K) | 2.27 (8K) 6.79 (8KCN) 6.82 (8KCNS) |

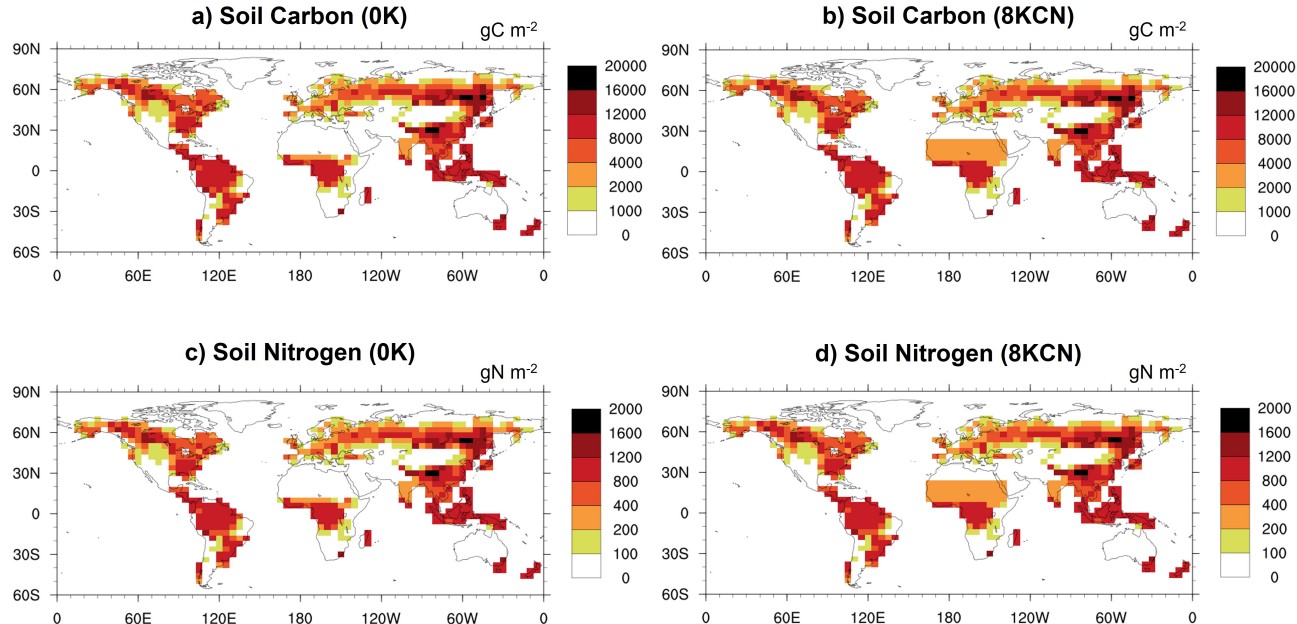

Figure 1: Global maps of soil carbon and nitrogen used in the 0K and 8KCN simulations. Values in boxed area is modified in the 8KCN experiment.

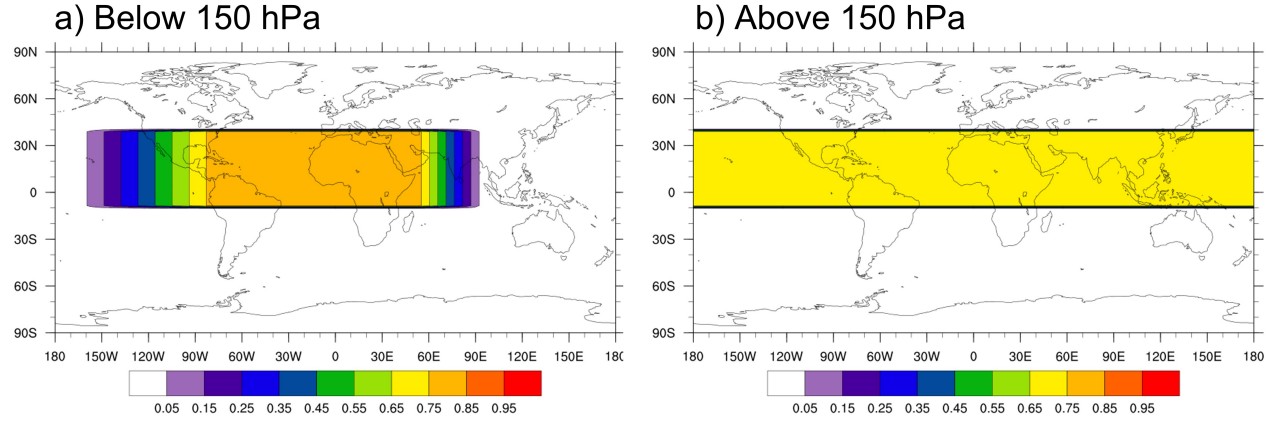

Figure 2: Dust reduction fraction a) below 150 hPa and b) above 150 hPa in the 8K experiments compared to the present–day values.

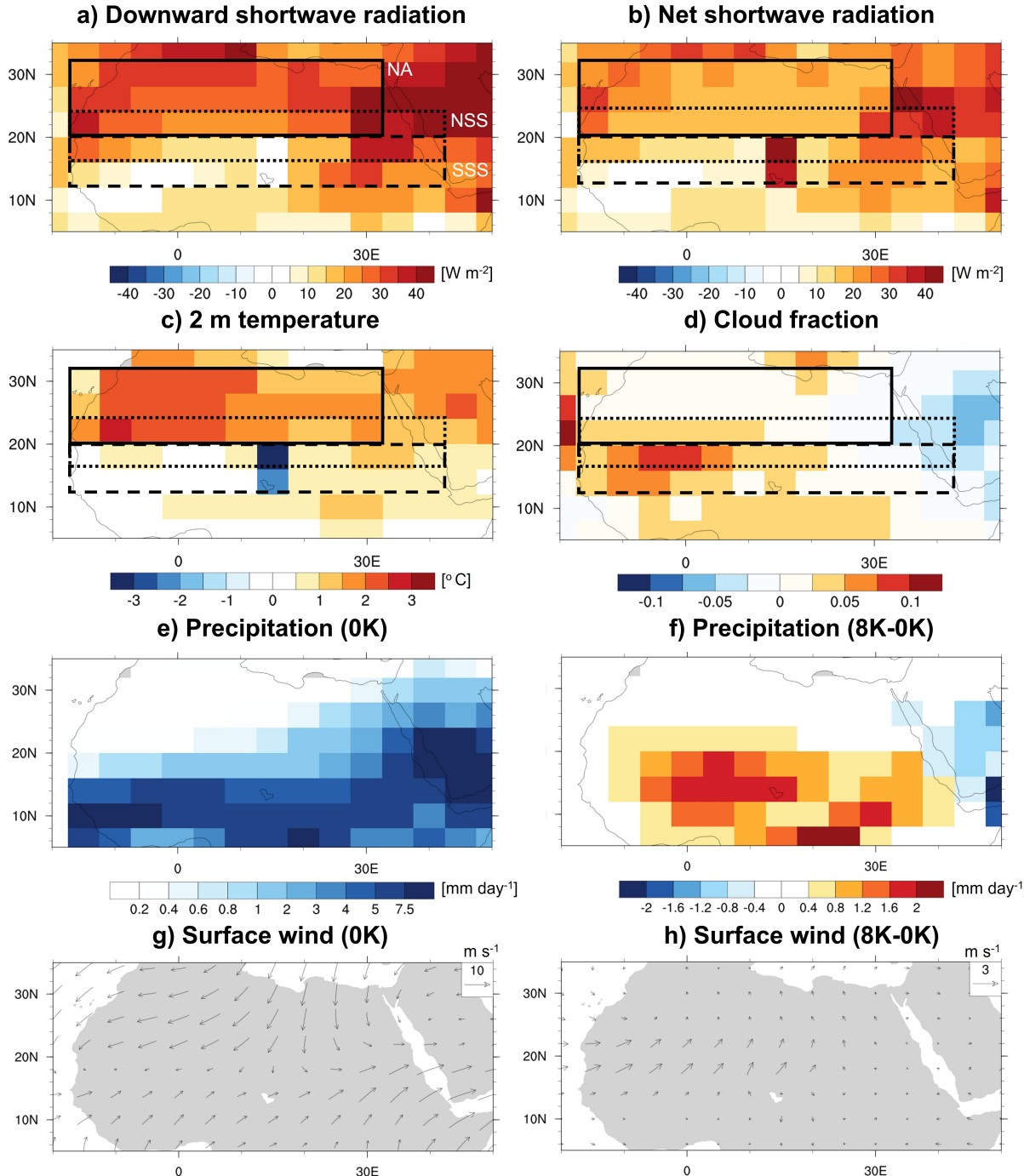

Figure 3: Differences of climate conditions between the 8K and 0K experiment (8K-0K) in summer (June, July, August, and September; JJAS). Differences in downward shortwave radiation (a), surface net shortwave radiation (b), 2 m air temperature (c), and cloud fraction (d). (e) and (f) are JJAS precipitation in the 0K and its differences from the 8K experiment and (g) and (h) are JJAS surface wind vector of 0K and 8K experiment respectively. Black squared boxes are the NA (solid), NSS (dotted), SSS (dashed) areas.

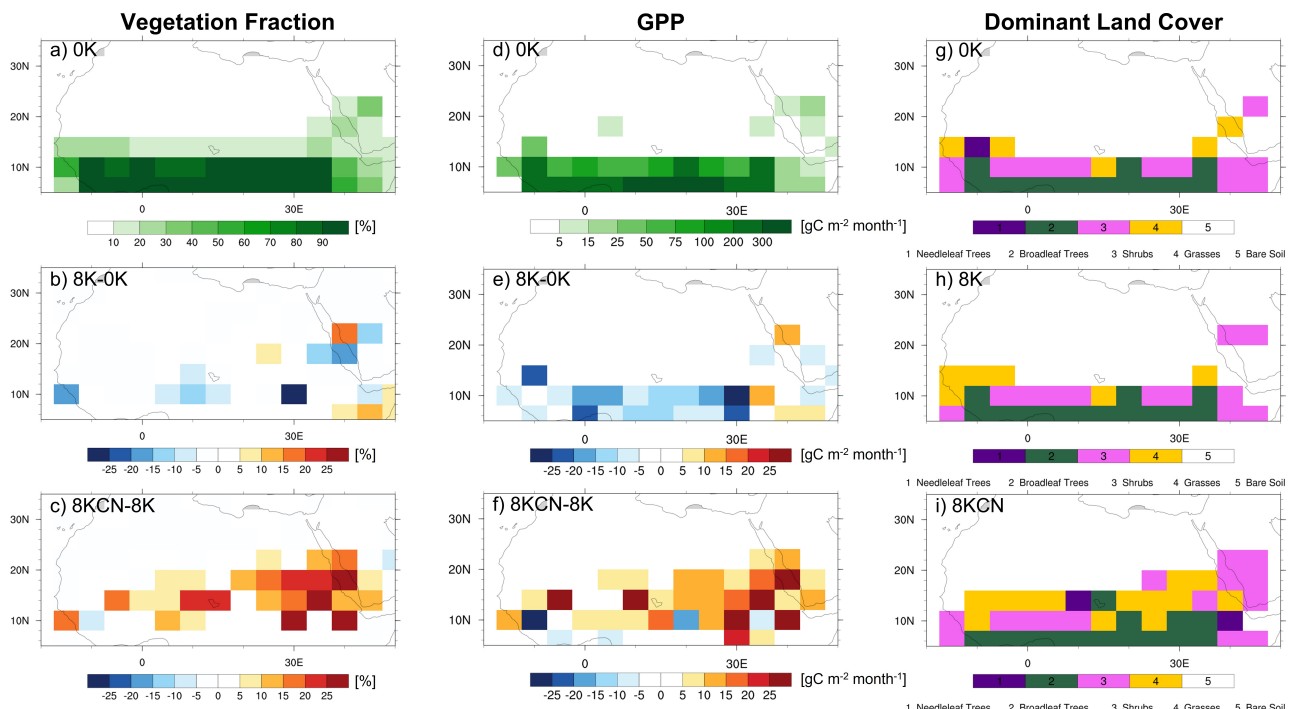

Figure 4: Vegetation fraction (percentage per grid, left column), summertime GPP (gC m$^{-2}$ month$^{-1}$, middle column), and dominant land cover (right column). Panels (b), (e) and (c), (f) are differences between 8K and 0K and between 8KCN and 8K, respectively. Dominant land cover is decided as bare soil if vegetation cover fraction is less than 20%.

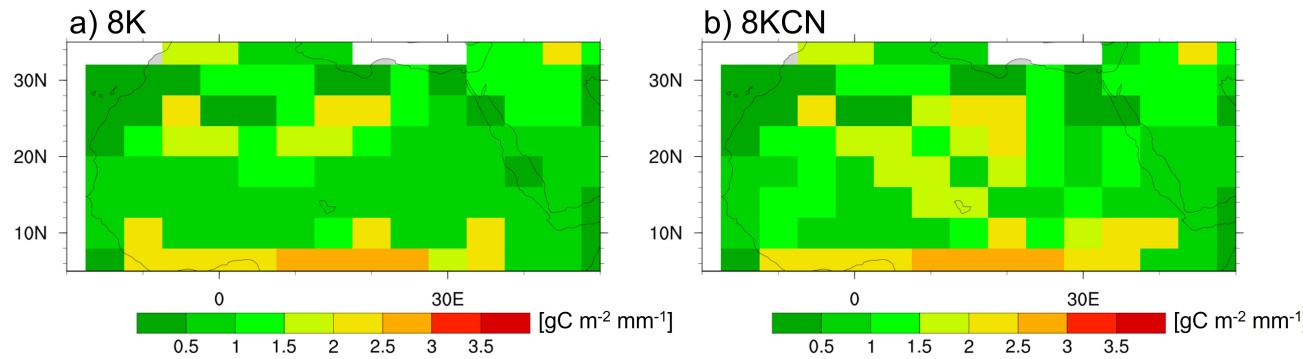

Figure 5: WUE (Water Use Efficiency, gC m$^{-2}$ mm$^{-1}$) during the growing season in the 8K (a) and 8KCN (b) simulations

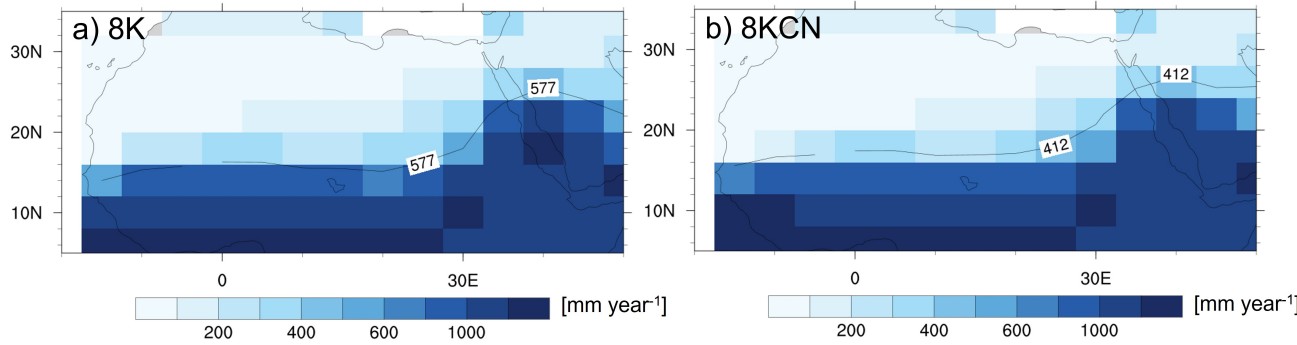

Figure 6: Annual precipitation in North African from a) the 8K b) 8KCN simulations. The minimum amount of precipitation for vegetation growth boundary is plotted as a black line with its values.

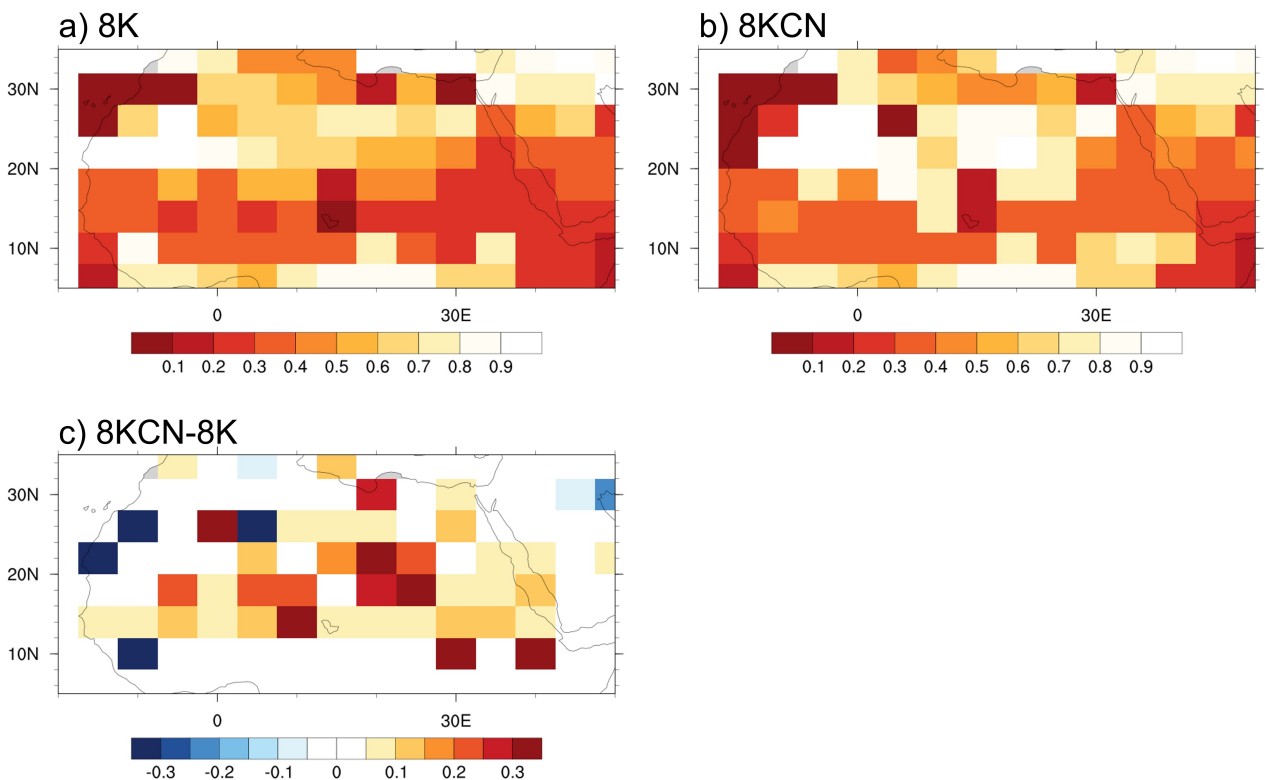

Figure 7: Downregulation fraction (GPP divided by GPP not limited by nitrogen) in the a) 8K and b) 8KCN simulations (top). Downregulation fraction difference between in the 8KCN and 8K simulation (bottom, Figure 7c).

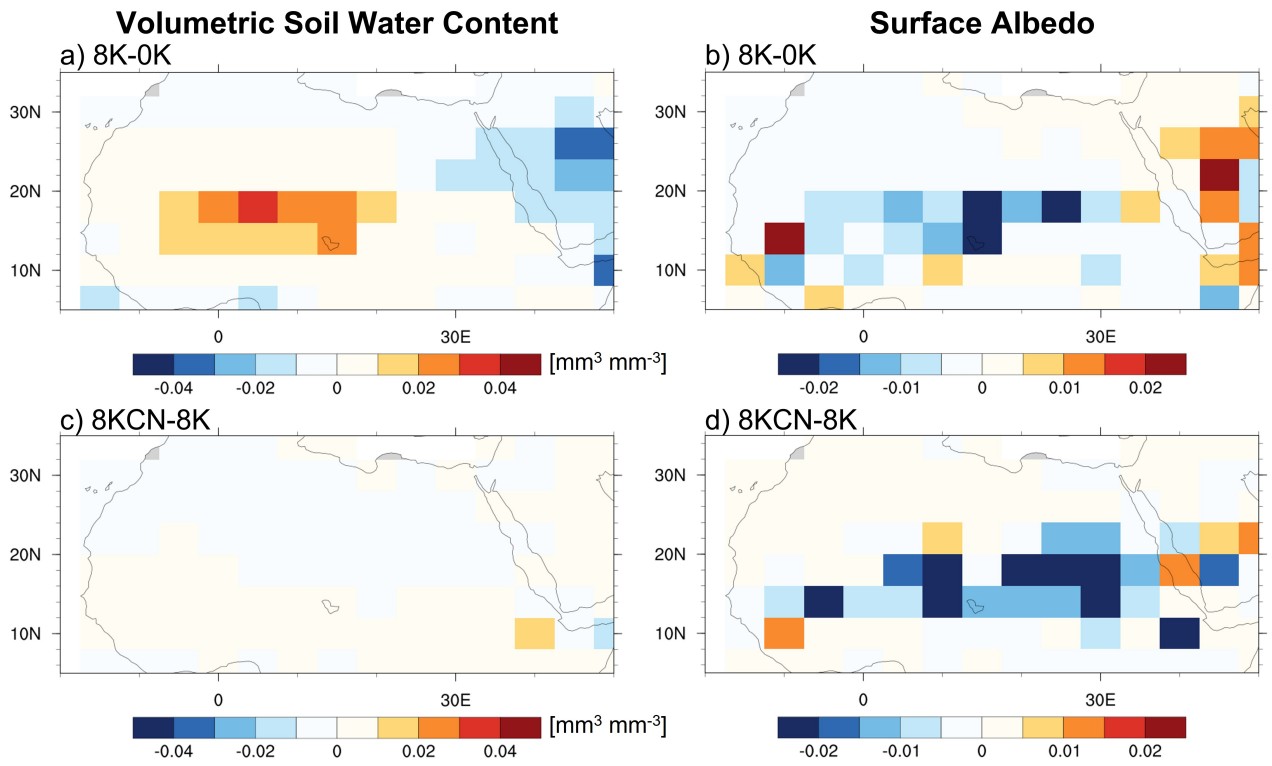

Figure 8: Differences in volumetric soil water content (left) and surface albedo (right) between the 0K, 8K, and 8KCN experiments. The
first and second rows show the differences between the 8K and 0K, between the 8KCN and 8Kexperiments, respectively.

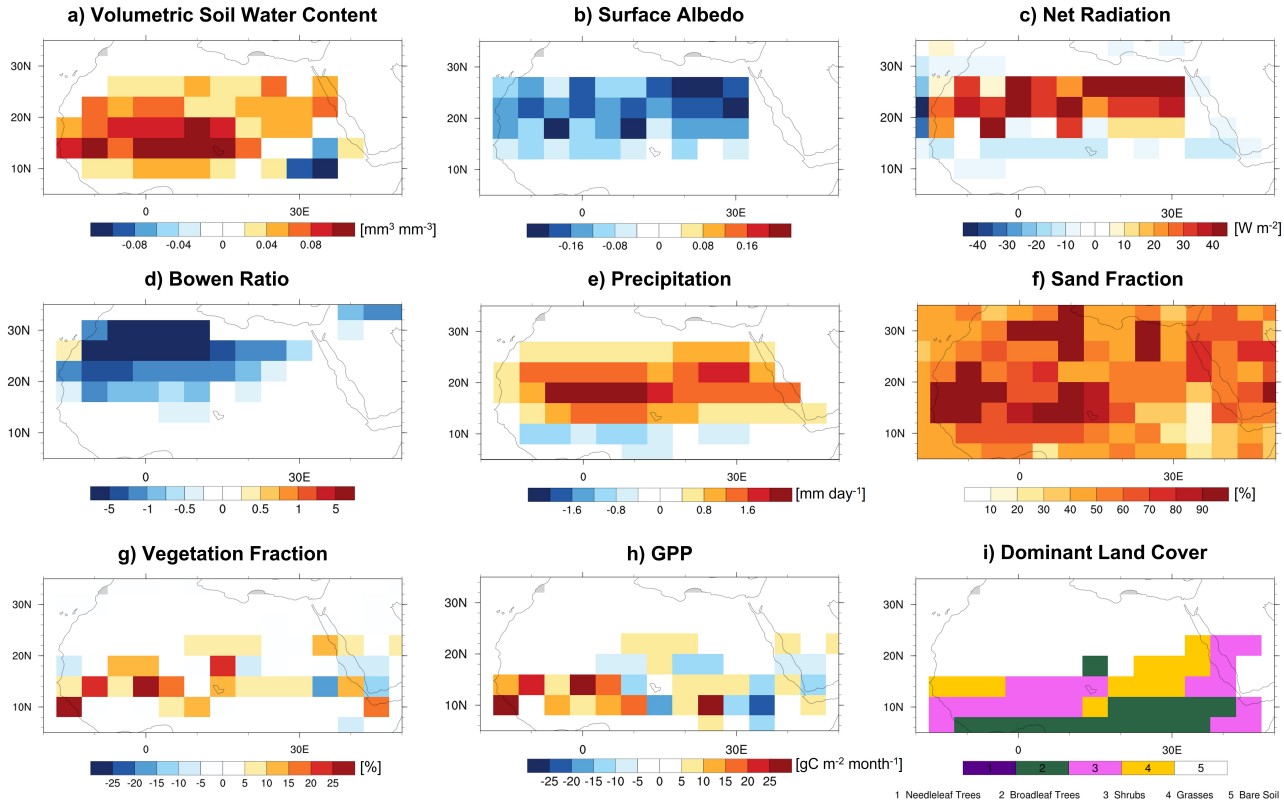

Figure 9: Differences of the loamy soil experiment (8KCNS) from 8KCN. (a) volumetric soil water content, (b) surface albedo, (c) net shortwave radiation, (d) Bowen ratio, (e) precipitation, (f) sand fraction in grid used in 8K and 8KCN, (g) evapotranspiration, (h) 600 hPa wind of 8KCN, (i) 600 hPa wind difference, (j) vegetation fraction, (k) GPP, and (l) dominant land cover in the 8KCNS experiment. Only grids that are significantly different at 95% in a Student's t test are colored in Figure 9e. Soil type is decided as the relative percentages of sand, silt and clay in soil and soil physical parameters are assigned based on this soil type in the CLM model (Figure 8f; Bonan et al., 2002).