# Peer review of "Figure S1: JJAS Precipitation (a) difference and (b) area of difference significant at 95% in a Student's t test, (c) JJAS surface temperature and (d) JJAS surface pressure difference for 8KCN and 8KCN0L simulation (8KCN – 8KCN0L). The black box is the grid of prescribed lakes."

_Climate of the Past, 2021_

## Author Comment (AC1)

Review of "Effect of nitrogen limitation and soil processes on mid–Holocene greening of the Sahara" by Lee et al.

In this work Lee et al. present a range of numerical modelling results aimed at investigating the effects of nitrogen limitation and soil processes on the vegetation expansion in North Africa during the mid-Holocene. This is an important question as the authors demonstrate they would have contributed significantly to the greening of the Sahara.

The authors used an advanced ESM, while the model configuration may have some flaws. Multiple global simulations of 8000 years ago were performed with different model set-up targeting different combinations of soil carbon, nitrogen, and lake area, etc. Some simulations require additional model input such as the mid-Holocene soil carbon and nitrogen in North Africa, and the authors have made plausible assumptions.

The results highlight the albedo- and evapotranspiration-precipitation feedbacks related to soil biophysical properties and soil nitrogen limitation, and they can potentially explain why current ESMs fail to realistically simulate a Green Sahara.

I think this is an interesting study and it is easy to understand its scientific message. However, I suggest some points, mostly regarding the methodologies, be addressed or be clarified before it is considered for publication, as follows.

1. I feel the MH period that is used in this study (8 ka) should be more clearly stated in the title or abstract. Since 8 ka is a less common model setup compared to 6 ka, some limited efforts would be beneficial to the modelling community to quantify the differences in climate/ecosystem response between 8 ka and 6 ka.

> Reply: We selected 8ka because previous studies reported that Sahara was greener in this period than 6ka (Jolly et al., 1998; Groner et al., 2018) as we already mentioned it in 2.2. We revised the abstract to clarify this selection of modeling period.

2. The term "Earth system model"  (widely used in the text) is confusing. As I read in Sec 2.2, there is neither description on the ocean/sea-ice model component that was employed, nor how the ocean state was prescribed. The authors might have used a model configuration of only dynamic atmosphere and land surface, which can cause serious caveats. Ocean dynamics plays an important role in amplifying the orbitally induced strengthening of WAM (e.g. Braconnot et al. 1999).

> Reply: As far as we understand, Earth system model (ESM) indicates as global climate models that incorporate global carbon cycle into the model. In this sense, CESM can be called earth system model and full name of CESM is the community ESM.

Please consider that we simulated the green Sahara (GS) using one of the ESMs with the pre-industrial ocean data with the nitrogen parameterizion. We fully agree that ocean dynamics are important in correct simulation of the GS and we already cite Branconnot et al. (1999) in our manuscript. However, please consider that our focus is to clarify the effect of nitrogen limitation and soil processes on the green Sahara simulation rather than exact reconstruction of the GS. We revised our manuscript for better readability of this issue.

3. The model spatial resolution (4 deg in N-S direction) is very coarse. For a range of the whole simulations, the vegetation extent only shifts a few gridcells (e.g. only one gridcell from extreme cases of 0K to 8KCNS, no shift from 0K to 8K or from 8KCN to 8KCNS). With such a coarse resolution, is the division of three boxes (the overlap takes up half of the box) necessary? Is it feasible that the authors repeat some representative simulations (e.g. 8K and 8KCNS) with higher resolution? Also, how can the authors estimate 2 deg shift (Line 192) from 4 deg model output?

> Reply: We fully understand this concern on the resolution. We think that benefits from high resolution simulation is not much than the current coarse simulation because we clarified the underlying physical mechanism to regulate the GS by nitrogen limitations and soil physical properties. We expect to get the same results in the higher resolution simulations because the same underlying processes should be even in the high resolution simulations, which is important contribution of our study. Please also consider that our computing power is not enough to catch up much higher resolution simulations of the CESM.

4. It is quite puzzling as the authors reported that the model does not simulate increased vegetation fraction or GPP even with increased precipitation in 8K (Fig. 4). Are the vegetation fraction, GPP and dominant land cover type annual mean or JJAS mean (not clearly stated for soil water, albedo, etc.)? Could there be a mismatch? Could it be related to the coarse spatial resolution? It is also not easy to understand when the dominant land cover type is unchanged as bare soil, the vegetation fraction and GPP changes over these gridcells can be quite large (e.g. Fig. 4 8K vs. 8KCN).

>> Reply: Please make sure that vegetation fraction does not increase in 8K because nitrogen limitation does not make substantial GPP to make extensive vegetation in this region, which is discussed in our manuscript. Also consider that in the 8K simulation, there is slight increase in vegetation in the area where GPP increases as the reviewer pointed out. However, such increases are relatively small and so

dominant land cover is still bare soil. This is not mismatch with any other related variables and please consider that our contributions are to find physical mechanism to make such simulation results based on code analysis and sensitivity experiments in our manuscript. This is why we believe that this cannot be explained by the model resolution because we explain physical mechanism on such unchanged vegetation fraction despite increases in precipitation in the 8K. We revised our manuscript to incorporate the comments for better readability.

5. It can be more informative for earth system modelers if the authors can do an estimate for the relative contribution from each process to the greening of North Africa. It should not be difficult if the authors create an index to quantify the vegetation condition within a box and assume all the factors (MH boundary conditions, soil texture, nitrogen limitation) can be added up linearly.

>> Reply: We revised our manuscript to incorporate the reviewer's comment by adding relative contributions of soil nitrogen, soil type, and orbital change to the greening of the Sahara in the SS region to section 3.3.

Technical:

L32 platform -> example

>> Reply: As the reviewer suggested, we changed the word.

L33 respects -> aspects

>> Reply: As the reviewer suggested, we changed the word.

L101 iss -> is

>> Reply: We corrected it. Thank you.

Fig. 8f Please explain more on the sand fraction used here.

>> Reply: Soil type is decided as the relative percentages of sand, silt, and clay in soil and soil physical parameters are assigned based on the soil type (Hurrell et al., 2013). We revised our texts to clarify the meaning of sand fraction in the figure.

---

## Author Comment (AC2)

l. 47/48: "Furthermore, vegetation in the GS led to more simulated organic matter in soils and therefore affecting hydraulic, thermal, and radiative properties of soil." => From the context, it's not entirely clear whether you are referring to our own study, or another previous study (which I assume you do). If the latter is the case, add the reference to make it clear that you are referring to work that has been conducted previously.

> Reply: We want to cite previous studies to attempt to understand soil influences on the vegetation in the GS and we revised our texts for better readability, as the reviewer pointed out. Thank you.

l. 48: soil texture changes: Be careful with terminology here and in other places in the manuscript: Texture is a quality that refers to the grain size distribution of the mineral soil component, usually represented by giving the mixing ratio between the grain size classes sand, silt, and clay. Texture normally does not change due to biological processes, only through geological processes such as weathering, erosion, and deposition/accumulation of material. Adding soil organic matter (SOM) to the soil therefore will not change texture, although it affects soil properties such as bulk density, aggregate formation, size and stability, and water holding capacity.

> Reply: We fully agree with the reviewer, and we revised our manuscript based on the reviewer pointed out.

l. 52-56: So this is in the first place a model calibration study? That's the impression one gets from reading this paragraph. It's rather vague. I think you should specifically state the research questions that you have regarding the GS and that you are attempting to address in this study. And, consecutively, later on, present your answers and conclusions regarding these research questions in the results and discussion section of the manuscript. It would help the reader to know what to expect, and where the focus of this study will be.

> Reply: We revised this paragraph as the reviewer suggested.

l. 66 "… and CFexess is the excess of carbon fluxes…" => Definition? What does it mean/what is the reference basis - Please verbally clarify the context of "excess" here so that it is more clear where you are heading to/what you are demonstrating with the following equations. E.g., "CFexcess is the amount of carbon that plants could additionally allocate if nitrogen were not limiting."

> Reply: We revised this sentence and its definition in the Appendix as the reviewer

suggested.

l. 70, Eqn. 3: kn_alloc/kc_alloc => Is the purpose of this term to aim for a target C/N ratio in plant tissue?

> Reply: This is not related to C/N ratio that our study investigated. We revised our texts, equations and appendix to clarify impacts of soil carbon and nitrogen on GPP regulation.

l. 70: Nretrans: Not clear to me what "retranslocated" is? Nitrogen re-allocated within the plant from one compartment to another? Please clarify.

Reply: "$N_{retrans}$" was defined in the Appendix. We added more information on this issue in this paragraph and in the Appendix for better readability.

l. 83: "surface albedo is irrelevant to soil moisture" => I guess you mean the other way around: "soil moisture is irrelevant for surface albedo..." or better "surface albedo is not influenced by soil moisture".

> Reply: We revised this sentence as the reviewer suggested.

l. 88, Eqn. 9: Delta becomes negative at theta1 > 0.275. What do you do at water contents > 27.5%? Assume the soil does not get darker any further than the value set for alpha_sat?

> Reply: We omitted some information in this equation, and we revised this equation for what the reviewer pointed out.

l. 89/90 What is the range of albedo values associated with these classes, i.e., what is the albedo of the lightest, and what is the albedo of the darkest soil? Just to get a feeling for the range. And: is this albedo for the classes equal to alpha_sat or the alpha of nearly completely dry soil?

> Reply: We added more information as the reviewer suggested. As written in the paper, there are 20 color classes in CLM. In the lowest albedo (darkest soil, class 1), the dry (saturated) albedo is 0.08 (0.04) and 0.16 (0.08) in visible and NIR range, respectively. In the highest albedo (lightest soil, class 20), the dry (saturated) albedo is 0.36 (0.25) and 0.61 (0.50) in visible and NIR range, respectively. The smallest albedo in the model, alpha_sat, equals to the saturated soil. We add this information to the manuscript for better readability.

l. 97: "The MH period is set to 8000 years ago (8K) in this study" => Terming it "Mid-Holocene" is then a bit misleading, because the Mid-Holocene normally is attributed to the time period between 7000-5000 BP, centering around 6000 BP (https://www.ncdc.noaa.gov/global-warming/mid-holocene-warm-period).

> Reply: We revised the word mid-Holocene (MH) to early to mid-holocene (EMH) as the reviewer suggested.

l. 101: "... soil texture change to loam iss added to the 8KCN in the 8KCNS" => see comment for line 48 – soil texture is unlikely to change from sand to loam, and definitely not because of more SOM. It will still remain sand, even if it is sand with a high content of organic material mixed in. The organic material does not add more silt and clay to the sand. As a sensitivity study, it is maybe okay to conduct this experiment, although I do not understand why the effect of SOM on soil bulk density, hydraulic capacity, and other soil properties is not directly accounted for as a process in the model. If it is not, then this should be stated, and explained that using loam is a surrogate to mimic the behavior of sandy soil with increased SOM.

> Reply: Two things are mixed up in this paragraph and we revised our texts for their clarification. The experiment to increase SOM (8KCN) needs to be differentiated from that to change soil type from sand to loam (8KCNS). Our study shows that it is not enough to increase SOM in sandy soil to incorporate vegetation impacts on soil in the GS. Soil in the GS contained more nutrients and humus because of extensive vegetations in this region, which indicates that soil type in the GS may have more loamy soil (Levis et al., 2004).

l. 103-106: "It is reasonable to consider more soil carbon and nitrogen in the GS than in the present-day Sahara, because of their close coupling with biomass..." => I'm not entirely sure I understand this correctly: did you entirely prescribe higher C and N content in the soils, or only as an initial condition, and then you let soil C and N evolve dynamically due to the vegetation and organic matter decomposition you simulated? Where vegetation is present, organic matter should automatically accumulate in the soil and lead to an increase in C and N over time?

> Reply: In our 8K experiments, vegetation does not increase despite increased precipitation in the EMH period because of the nitrogen limitation in the less soil organic matter and sandy soil. This does not increase soil C and N and leads to less vegetation subsequently. That is, GPP is not enough to make extensive green vegetation. To consider this feedback between vegetation and soil, larger soil C and

N is assigned in our 8KCN experiment as initial values and then they evolve dynamically through litter-fall, decomposition, and decaying processes in the model. We revised our texts to make this point clearer.

l. 111-113: Short justification why these three areas are critical/what makes them critical? Are they particularly sensitive to tipping behavior, and if yes, why? Do they have a special role for feedback in the climate system?

> Reply: It has been reported that the SS region was sensitive to precipitation-land interaction and the NA region is closely related to the onset and progress of the African Monsoon (Kutzbach et al., 1996). We added this information into the texts.

l. 114 I'm missing section 3, and stumbled when reading because I first had to notice that this is now the results section. Please add a formal section "3. Results", with a brief general introduction to your results.

> Reply: We revised our texts as the reviewer suggested.

Figure 3: Maybe this is just a personal matter, but for the wind field difference in panel h), I find it a bit difficult to grasp the difference and mentally translate it back into what the actual wind field would have looked like. Here I'd find it easier if you showed the actual 8K wind field next to the 0K wind field. At least for me, it's easier to see the difference between both absolute fields than to re-translate the difference into the actual field.

> Reply: We revised this figure as the reviewer suggested.

l. 117 "increases" => Same as for methods section: report results in the past tense. They won't change anymore.

> Reply: We are not native speakers, but we feel that present tense is possible in this part like many other papers for vivid expression. Also, English proof reading by a native speaker did not give any correction on this. Accordingly, we want to use the present tense if you don't mind it.

l. 122 "the increase in air temperature is not substantial" => What does that mean in more concrete terms? Statistically non-significant?

> Reply: We used this sentence when temperature change is less than 10 % and we added more information to this paragraph.

l. 138 ff: The precipitation minimum allowing vegetation growth is excessively high at 577 mm/year! This is the amount of precipitation typical for more arid savannas, not even grasslands. If the model does not manage to produce any substantial vegetation below this threshold, I see a fundamental issue there that quite heavily impacts the meaning of this study. You make nitrogen availability, or limitation thereof, responsible for the ability of the model to simulate the GS or fail to do so. But how can you be sure it is the nitrogen availability if, in fact, the water-vegetation linkage is so strongly off? In my opinion, this is not only an uncertainty that has not been clearly discussed in previous studies, it is rather an impediment to the aim and scope of your study. In addition, I wonder why nobody has looked into that problem and tried to find the reason behind it if it has obviously been noticed in previous studies. It seems to me that this problem needs urgent fixing. Have you ever checked on water use efficiency (WUE, annual GPP/annual transpiration per unit area)? I suspect that WUE is likely way off compared to remote sensing benchmarks. Or alternatively, that the SPA continuum (soil-plant-atmosphere continuum) is somehow poorly represented/broken. Or that drainage/runoff is too high so that the water is gone before it becomes accessible for the plants. In any case, even if it is not possible for now to fix this issue, it should be discussed in detail with regard to its implications for the current study, and what the study can reveal in the light of this deficiency in the model.

> Reply: We fully agree that this should have high priority in the future research. Please note that most of LSMs have much larger threshold values compared to the observation and this was recently reported by Hopcroft et al. (2017). Our study shows that 1) the model cannot make substantial GPP for extensive vegetation cover because of the nitrogen limitation by the model code analysis and the sensitivity analysis, 2) these large threshold becomes smaller if you properly assign soil organic matter and soil type, thus indicating the WUE increases. In reality, WUE increases if we consider the SOM and soil type properly. We believe that our findings are important for climate simulation especially with the nitrogen cycle and are the first report as far as we know. However, we still don't know direct linkages exactly with this issue. In these perspectives, we revised our manuscript by mentioning this issue and limits of our study which remains one of

l. 157 "...results from less downregulation due to..." => "... results from reduced N-limitation on photosynthetic C-gain and GPP... "

> Reply: As the reviewer suggested, we re-wrote our sentence.

Fig. 6: These are both 8k results in reference to results without nitrogen limitation, so there must have been an 8k run that had no nitrogen limitation? This is the first time such a control is suggested (one may guess it must have been done based on the figure). If so, it should be stated more clearly in the methods section and added as a control scenario in Table 1. Figure caption: I'd rather call that "fractional GPP reduction" than "downregulation fraction". And I'd personally reverse the color scheme (higher fractions mean less reduction means lighter color?). In addition, an additional diff-map between a) and b) may show more clearly where and by how much GPP was enhanced due to the additional nitrogen in the 8KCN scenario (for example, b divided by a, or b-a).

> Reply: As the reviewer suggested, we revised our texts and figure.

l. 173 "... the soils in the GS were loamier because of the larger organic matter in soil,..." => As stated earlier: Loamyness is NOT defined by soil organic matter content. It depends on the mixture ratio between sand, silt and clay, i.e., the inorganic component of the soil. And that was very likely not much different from today. Being richer in SOM does NOT mean loamier. However, it does affect soil bulk density, aggregate formation, hydraulic and thermal properties, and probably also albedo. So in that regard, your assumptions regarding the sensitivity study are valid. Just change the "loamy" part to not tie these changes to texture. Or did you have no other way to mimic the changes in soil properties caused by more SOM than using loamy texture as a surrogate? In this case, it should be explicitly stated and discussed with regard to its validity.

> Reply: We think that this comment is related to other comment above and we revised our manuscript based on our reply above for your earlier comments.

l. 177 "... leads to significant changes in vegetation and climate..." => Statistically significant at what level? And is it significant for all your subareas, or only specific ones?

> Reply: We added t-test results and revised our manuscript based on the reviewer's comments.

l. 179: "a change from sandy to loamy soil leads to an increase in soil porosity" => The difference in porosity is comparably small (0.437 for sand and 0.463 for loam according to USDA soil texture classification). What matters more is the difference in saturated hydraulic conductivity, which is more than one order of magnitude lower for loamy soil compared to sandy soil (5.040 m/day for sand, 0.317 m/day for loam), which implies that water drains way more slowly from a loamy soil as opposed to

sandy soil. (see Tab. 3 in DOI 10.1007/s11269-013-0295-2)

> Reply: We revised our manuscript to incorporate this reviewer's comment.

l. 180: "These changes lead to an increase in net radiation" => maybe rephrase? "... led to enhanced absorption of radiation..."?

> Reply : We rephrased our sentence as the reviewer suggested.

l. 184 Rephrase. This sentence is hard to read.

> Reply : As the reviewer suggested, we rephrased this sentence to make it easier to read.

l. 188 ff. I'm not so sure that this is the actual cause behind the Sahara greening. I see a major problem with the model not capturing the vegetation-precipitation relationship in the first place. If the model does not simulate savanna or grassland at ca. 500 mm annual precipitation, then in my opinion this has a far larger effect than the effect caused by nitrogen limitation. Moreover, more vegetation due to more precipitation is required to increase soil nitrogen content compared to non-vegetated state due to N-fixation and N-input and accumulation in the soil as a consequence of biomass decomposition and turnover. Vegetation and N availability therefore can be expected to have built up correlated with each other.

> Reply: Please check our responses to your earlier comment on L138. Please note that even nitrogen limitation results in increases in vegetation if we properly assign the SOM and soil type with increases in precipitation. We believe that our sensitivity analysis shows relationship of the GS greening with N availability which is related to the SOM in the model. We tried to revise our manuscript to emphasize our conclusion clearly.

l. 195 general note:

Entirely missing: a discussion section with an in-depth discussion of the results and putting them in the broader context of other studies conducted on the GS-topic, both simulation studies, and proxy-based studies. Also no discussion of limitations of the current study, e.g., the poor representation of precipitation-vegetation cover linkage and its implications. This ought to be addressed.

Too many details are missing in this paper the way it is currently written so that it is in part hard to read without having to guess on background information. For example, it is not clear whether the 8K simulation had more or less nitrogen than the 8KCN simulation, i.e., whether that simulation was with or without nitrogen cycling considered. And if nitrogen cycling was considered in the 8K simulation, then it's not clear whether there was also a baseline simulation for 8K that had no N-limitation (N-cycling turned off), as may be inferred from Fig. 6., which must have a reference base that is not clear.

Reply: We revised our manuscript for better readability based on the reviewer's comment. Comments here are related to other comments mentioned above and please check our responses to other comments above. Please also consider that we discussed the physical mechanism why vegetation is not well simulated in the 8K experiment and suggested a remedy to better capture vegetation cover for the GS. Please also make sure that vegetation cannot exist despite ample precipitation in the soil and nitrogen parameterizations currently used in the model. We also suggest that this will be a significant issue for the future climate simulation and irreversible process detection. Generally, it has been reported that land surface processes can amplify the green Saharan vegetation with the orbital differences. Especially, it is known recently that the soil processes can be a key as in Lu et al., (2018). Our study extends that soil type and nitrogen parameterization, which was added into the state-of-the art earth system model, are critical factors for the GS simulations. As far as we know, this is the first study to find that soil nitrogen processes have a big impact on the Holocene greening. Nevertheless, we agree that there still were land processes that we cannot fully understand (e.g. the high precipitation threshold for vegetation growth) and we added sentences to bring up limitations of our study.

l. 208 "interactive changes in soil texture" => I'd rather say: "process-based dynamics of soil properties"

Reply: As the reviewer suggested, we revised the texts.

l. 208/209 "Notably, our findings and their implications can be extended to the future climate simulations" => How exactly? This is rather vague and general, it would be nice to have more details.

Reply: As the reviewer suggested, we revised the sentence by suggesting implications of our findings for climate modeling and future irreversible processes

General note: "Mega Lake Chad" => to my knowledge, officially it's called "Lake Mega-Chad"?

Reply: As the reviewer suggested, we changed the texts.

**Minor corrections/technical remarks:**

l. 31/32: "Our future climate prediction is made…" => "Future climate predictions are made…" - otherwise it reads like you are trying to make future climate predictions in this study, which is misleading.

>> Reply: As the reviewer suggested, we revised the texts.

l. 32: "In these respects" => "In this context"

>> Reply: As two reviewers suggested, we revised the texts.

l. 35: "Many modeling studies have been tried…" => "Many modeling studies have tried…"

>> Reply: As the reviewer suggested, we revised the texts.

l. 45/46: "… global carbon budget has been better captured by its downregulation effect of terrestrial GPP…" => "…the representation of the global carbon cycle has improved due to accounting for the N-limitation effect on GPP…"l. 47 "… and therefore affecting hydraulic…" => "…, which affected hydraulic,…"

Reply : We rephrased the sentences for better readability.

l. 80 "… is listed in Supplement" => "… are listed in the Supplement"

>> Reply: As the reviewer suggested, we revised the texts.

l. 99/100: "…examine the impacts soil nitrogen,…" => "examine the impacts of soil nitrogen,…"

>> Reply: As the reviewer suggested, we revised the texts.

l. 101: "…loam iss added…" => "… loam is added…'"

>> Reply: As the reviewer suggested, we revised the texts.

l. 117 "…than the present…" => "… compared to the present…"

>> Reply: As the reviewer suggested, we revised the texts.

l. 120 "This intensified land-sea thermal contrast yield spatial changes…" => "This intensified land-sea thermal contrast caused spatial changes…"

>> Reply: As the reviewer suggested, we revised the texts.

l. 122 "… because the increase in…" => "…because of the increase in…"

>> Reply: As the reviewer suggested, we revised the texts.

l. 123 "meridional wind" => "the meridional wind"

>> Reply: As the reviewer suggested, we revised the texts.

l. 125 "This ITCZ shift made a favorable condition…" => "This ITCZ shift caused a favorable condition…"

>> Reply: As the reviewer suggested, we revised the texts.

l. 126 "…in both of the…" => "… to both the…"

>> Reply: As the reviewer suggested, we revised the texts.

l. 134/135 "…extended more up north to…" => "… extended as far north as…"

>> Reply: As the reviewer suggested, we revised the texts.

l. 135 "…to proxy data…" => "…to the proxy data…"

>> Reply: As the reviewer suggested, we revised the texts.

l. 136 "… in western Africa and southern border…" => "… in western Africa and the southern border…"

>> Reply: As the reviewer suggested, we revised the texts.

l. 145 "… that Mega-Lage Chad does not make substantial changes…" => "… that Lake Mega-Chad did not cause substantial changes…"

>> Reply: As the reviewer suggested, we revised the texts.

l. 159 "increases about by" => "increases by about", or "increases ca."

>> Reply: As the reviewer suggested, we revised the texts.

l. 182/183 "in the North Africa" => "in North Africa", or "in northern Africa", or "in the north of Africa"

>> Reply: As the reviewer suggested, we revised the texts.

l. 136 "… that the evapotranspiration increase in the Sahara-Sahel region made an increase…" => "… that the evapotranspiration increase in the Sahara-Sahel region caused an increase…"

>> Reply: As the reviewer suggested, we revised the texts.

l. 188/198 "…vegetation change increases precipitation with enhanced evapotranspiration…" => due to enhanced evapotranspiration

>> Reply: As the reviewer suggested, we revised the texts.

l. 203 "thus making vegetation cover and GPP increases" => "thus making vegetation cover and GPP increase"

>> Reply: As the reviewer suggested, we revised the texts.

l. 205: "through the albedo-precipitation." => "through the albedo-precipitation feedback."

>> Reply: As the reviewer suggested, we revised the texts. Thank you very much for your constructive comments.

---

## Author Response (AR2)

To Dr. Qiuzhen Yin

*Climate of the Past*

Subject: Revision of manuscript entitled "Effect of nitrogen limitation and soil processes on Holocene greening of the Sahara"

Dear the editor of *Climate of the Past*

We appreciate the constructive comments of the editor and revised the manuscript by incorporating all of the comments provided by the editor. Below are our responses to the editor.

We believe that our findings will be of interest to corresponding science communities and please do not hesitate to contact with us for your further inquiries.

Thank you very much for your support.

Sincerely,

Jinkyu Hong

Department of Atmospheric Sciences, Yonsei University, Seoul, South Korea

Email: jhong@yonsei.ac.kr / hong.jinkyu@gmail.com

Fax: 82-2-2123-5693

Tel: 82-2-365-5163

1. Page 1, line 16: change it to "...simulate the Green Sahara at 8,000 years ago by using ..."; Change "earth" to "Earth".

> Reply: We changed the word as the editor suggested.

2. Reviewer 2 suggests to add discussion on the limitations of the current study, e.g. the poor representation of precipitation-vegetation cover linkage and its implications. I don't find it in the revised manuscript. Please add such discussion.

> Reply: We added relevant sentence in Summary and Conclusion as the editor suggested.

3. Soil texture and soil processes include many properties and processes which could interact with climate and vegetation, such as those discussed in Finke et al (2017, Geology) and Ranathunga et al (2021, Quaternary International). As far as I can see, the only change in your 8KCNS experiment as compared to 8KCN is that the North African soil is prescribed as loam. This is fine, but I would suggest to mention ignoring other soil properties and processes as one of the limitations in your current study. In the same line, I would suggest to be more specific in the title and abstract of your paper, because the wording "soil processes" "soil texture" "changes in soil properties" "soil biochemical and physical properties" are very general but actually only a few limited properties and processes are considered in your study.

> Reply: Please let us explain this issue. In typical climate models, many parameters for soil thermal and biogeochemical properties (e.g., porosity, hydraulic conductivity, thermal conductivity, heat capacity, etc) are assigned with soil types (e.g., loam, sand, clay, etc.). For example, if we change the soil type from sandy to loamy soil, all soil-related parameters change and are used in the modeling. In this sense, we may be able to use general terms like soil processes and soil properties and we specify these processes and properties in the abstract and summary. We also agree that there are even many processes that are not parameterized in the climate models and we mentioned this issues with the limit of our study. Thank you for your constructive comments.

4. In the 8KCNS experiment, is there feedback between vegetation and soil texture? If not, is it appropriate to call it "fully coupled simulations"(page 1, line 19)?

> Reply: Our intention to use "fully coupled" was the coupling between the atmosphere

model and vegetation model. But we agree that this can be misleading and so we replaced "fully coupled simulations" with "coupled simulations". Thank you.

5. A "data availability" section is required for the papers published in Climate of the Past. Please add such a section.

> Reply: We added information on data availability section as the editor suggested.

6. Table 1, row 3: change it to "…using the orbital parameters and CO2 concentration at 8,000 years ago (Table 2).". last row: "prescribed"

> Reply: We changed the sentence as the editor suggested.

7. Table 5: Please indicate the simulation dates of other models. 6K, 8K, 9K ?

> Reply: We indicated the simulation dates in the model as the editor suggested.

8. The quality of the figures needs to be improved. The lines and texts are not very sharp in the figures. Some gray lines occur around the panels in many figures.

> Reply: As the editor suggested, we updated all the figures.

9. Fig2 caption: change "the early to mid-Holocene" to "8K"

> Reply: We changed the word as the editor suggested.

10. Fig3: please explain in the paper why NSS and SSS share a common latitudinal band.

> Reply: We explained this common band in section 2.2 as the editor suggested.

11. indicate the unit of vegetation fraction in Fig4 caption and the unit of WUE in Fig5.

> Reply: We added the unit in both Figure 4 and Figure 5 as the editor suggested.